# Dietary Berberine and Ellagic Acid Supplementation Improve Growth Performance and Intestinal Damage by Regulating the Structural Function of Gut Microbiota and SCFAs in Weaned Piglets

**DOI:** 10.3390/microorganisms11051254

**Published:** 2023-05-10

**Authors:** Wenxia Qin, Zhendong Yu, Zhechang Li, Hengfeng Liu, Wei Li, Jianan Zhao, Yin Ren, Libao Ma

**Affiliations:** College of Animal Sciences and Technology, Huazhong Agricultural University, Wuhan 430070, China

**Keywords:** Berberine, ellagic acid, gut microbiota, SCFAs, weaned piglets

## Abstract

Early weaning is an effective method for improving the utilization rate of sows in intensive pig farms. However, weaning stress induces diarrhea and intestinal damage in piglets. Berberine (BBR) is known for its anti-diarrhea properties and ellagic acid (EA) is known for its antioxidant properties, however, whether their combination improves diarrhea and intestinal damage in piglets has not been studied, and the mechanism remains unclear. To explore the combined effects in this experiment, a total of 63 weaned piglets (Landrace × Yorkshire) were divided into three groups at 21 days. Piglets in the Ctrl group were treated with a basal diet and 2 mL saline orally, while those in the BE group were treated with a basal diet supplemented with 10 mg/kg (BW) BBR, 10 mg/kg (BW) EA, and 2 mL saline orally. Piglets in the FBE group were treated with a basal diet and 2 mL fecal microbiota suspension from the BE group orally, respectively, for 14 days. Compared with the Ctrl group, dietary supplementation with BE improved growth performance by increasing the average daily gain and average daily food intake and reducing the fecal score in weaned piglets. Dietary supplementation with BE also improved intestinal morphology and cell apoptosis by increasing the ratio of villus height to crypt depth and decreasing the average optical density of apoptotic cells; meanwhile, improvements also involved attenuating oxidative stress and intestinal barrier dysfunction by increasing the total antioxidant capacity, glutathione, and catalase, and upregulating the mRNA expressions of Occludin, Claudin-1, and ZO-1. Interestingly, the oral administration of a fecal microbiota suspension to piglets fed BE had similar effects to those of the BE group. According to 16S rDNA sequencing analysis, dietary supplementation with BE altered the composition of the microbiota, including *firmicutes*, *bacteroidetes*, *lactobacillus*, *phascolarctobacterium*, and *parabacteroides*, and increased the metabolites of propionate and butyrate. In addition, Spearman analysis revealed that improvements in growth performance and intestinal damage were significantly correlated with differential bacteria and short-chain fatty acids (SCFAs). In brief, dietary supplementation with BE improved the growth performance and intestinal damage by altering the gut microbiota composition and SCFAs in weaned piglets.

## 1. Introduction

Early weaning at 3 weeks of age is a common technique on intensive pig farms, whereas the natural weaning time of piglets is approximately 17 weeks. However, early weaning can cause oxidative stress and dysbiosis of the gut microflora [1,2,3], which are essential factors in the etiology of intestinal damage and diarrhea in piglets [4,5]. In addition, with the comprehensive prohibition of antibiotics, it is critical to find suitable and effective antibiotic substitutes to treat a series of adverse reactions caused by the early weaning of piglets.

Previous studies have shown that plant polyphenol extracts play an important role in replacing antibiotics for the treatment of oxidative stress, intestinal damage, and diarrhea in weaned piglets [6,7,8]. Berberine (BBR) and ellagic acid (EA) have attracted considerable attention in recent years because of their extensive pharmacological functions. BBR is an isoquinoline alkaloid mainly found in pharmacological plants, such as *Berberis vulgaris*, *Coptis chinensis*, and *Phellodendron*, with low pharmacological toxicity [9,10,11] and has been extensively used as a common prescription drug to medically treat various bacteria-associated issues [12]. A study has shown that BBR prolongs the time leading up to diarrhea in a dose-dependent manner and improves diarrhea-predominant irritable bowel syndrome (IBS-D) symptoms in mice. BBR also enhances the barrier function of porcine intestinal epithelial cells [13,14]. However, there is little evidence indicating that dietary supplementation with BBR can reduce diarrhea in early-weaned piglets. Oxidative stress is an essential factor in the pathogenesis of intestinal mucosal damage [4], and EA is a polyphenolic compound containing four hydroxyl groups, extensively found in gallnuts, blackberries, and some vegetables, with strong anti-inflammatory and antioxidant activities [15,16]. A previous study shows that dietary supplementation with EA ameliorates the paraquat-induced oxidative stress by activating the nuclear factor erythroid 2-related factor 2 (Nrf2) signaling pathway in piglets [17]. Qi et al. [18] also found that EA significantly decreases the occurrence of paraquat-induced liver injury and inflammation in piglets. Moreover, a large amount of evidence clearly reveals that combinations of several active substances have greater advantages than individual substances [19], and clinical results have shown that the combination of *Coptis chinensis* and gallnut has a stronger antibacterial effect than either substance alone [20]. Based on the above studies, and because of the strong anti-diarrhea ability of BBR and the strong antioxidant ability of EA, we wondered whether the combination of BBR and EA could ameliorate diarrhea and intestinal damage in early-weaned piglets.

Recent research suggests that natural plant polyphenol extracts, such as resveratrol and quercetin, improve antioxidant capacity and alleviate intestinal damage in piglets by regulating the gut microbiota [21,22]. Similarly, BBR treats emergent diarrhea in mice by increasing the abundance of *Prevotellaceae_UCG-001* [23], and EA alleviates oxidative stress and liver injury in mice by altering the abundance of *Bifidobacterium* and *Alistipes* at the genus level [24]. Qin et al. also showed that *Akkermansia muciniphila* and *Veillonella parvula* are positively correlated with antioxidant activity in weaned piglets treated with EA [25]. However, it is worth noting that there is still no study showing whether the combination of BBR and EA can attenuate diarrhea and intestinal damage by modulating the gut microbiota composition in early-weaned piglets. Fecal microbiota transplantation (FMT) is the infusion of fecal filtrate from a donor into a recipient [26], providing an effective method to reveal the role of gut microbiota in the efficacy of natural plant polyphenols [27]. To test our hypothesis, we treated weaned piglets with BBR and EA and transferred their fecal microbiota to recipient piglets.

## 2. Materials and Methods

### 2.1. Experimental Design

We allotted 63 weaned piglets (Landrace × Yorkshire) at 21 days to 3 groups (21 piglets per group, each group was divided into 3 pens with 7 piglets per pen, the temperature of each pen was 28–32 °C, and the humidity was 60~80%). Piglets in the Ctrl group were treated with a basal diet and 2 mL saline orally, while those in the BE group were treated with a basal diet supplemented with 10 mg/kg (BW) BBR, 10 mg/kg (BW) EA, and 2 mL saline orally. Piglets in the FBE group were treated with a basal diet and 2 mL fecal microbiota suspension from the BE group orally. Each treatment lasted 14 days. A total of 7 piglets weighing 6 ± 1 kg from each group were selected for slaughter (n = 7). During the experiment, weaned piglets in each group were free to eat and drink, and the basic diet (Table 1) met the nutritional requirements recommended by the National Research Council (NRC, 2012). The BBR and EA were obtained from Sichuan Hengrui-Tongda Biotechnology Co., Ltd. (Chengdu, China) and Hubei Tianxin Biotechnology Co., Ltd. (Wuhan, China), respectively.

### 2.2. Sample Collection and Processing

The blood was collected from the vena cava and placed in anticoagulant tubes, which were stood at room temperature for 30 min and centrifuged at 3000× *g* for 15 min; the serum was placed in a 1.5 mL centrifuge tube and stored at −20 °C. A total of 3 3-cm long segments of jejunum tissue were sampled from the middle portion of the small intestine and gently rinsed with phosphate buffer saline (PBS), the upper part of the jejunum was placed in 4% paraformaldehyde (Thermo Fisher, Shanghai, China) solution for preparing intestinal tissue sections, while the others were quick-frozen with liquid nitrogen and stored at −80 °C. In addition, jejunal and colonic contents were taken from the middle section and transferred to 15 mL sterile centrifuge tubes, quick-frozen with liquid nitrogen, and stored at −80 °C.

### 2.3. Fecal Microbiota Transplantation

Stools were collected from 21 piglets in the BE group, homogenized with sterile normal saline, filtered through a sterile gauze and a 0.224 mm stainless steel tank filter, and centrifuged at 3500× *g* for 10 min. Fecal sediment was added to sterile normal saline to form a slurry, and optical microscopy was used to count the number of viable bacteria. Subsequently, sterile glycerol was added to the slurry at a final concentration of 10% and the number of viable bacteria was 10^8^ CFU/mL [28]. Finally, the fecal slurry was stored at −80 °C. Each piglet in the FBE group was orally administrated 2 mL fecal slurry every other day for 14 days.

### 2.4. Growth Performance and Fecal Score

The growth performance of the piglets was presented as average daily gain (ADG), average daily food intake (ADFI), and diarrhea rate. The ADG of the piglets in each group during the 14-day experiment was calculated by recording the body weight at the beginning and end of the experiment. The ADFI of piglets during the 14-day experiment was calculated by recording the daily and residual feed. The method of fecal scoring was as follows: normal feces (solid) scored 1, wet feces (semi-solid) scored 2, mild diarrhea (mushy) scored 3, and severe diarrhea (water sample) scored 4 points [22]. Mild and severe diarrhea were rated as occurrences of diarrhea.

### 2.5. Jejunum Tissue Morphology and Apoptosis of Epithelial Cells Examination

The upper part of the jejunum was immersed in 4% paraformaldehyde (Thermo Fisher, Shanghai, China) and stained with Hematoxylin-eosin (HE) (Sinopharm, Shanghai, China). The stained sections were observed using an Olympus BX51 microscope (Olympus, Tokyo, Japan) with an integrated digital imaging analysis system, and t villus height and crypt depth were measured using ImageJ software (version 1.8.0). In addition, terminal deoxynucleotide transferase d-UTP terminal labeling (TUNEL) was used to label the apoptosis of jejunum epithelial cells; the apoptotic cells were marked with green and the average optical density (AOD) was calculated using Image J.

### 2.6. Antioxidant Capacity Determination

The antioxidant indices of the middle part of the jejunal tissue were measured using kits obtained from Solarbio Technology Co., Ltd. (Beijing, China), which mainly included total antioxidant capacity (T-AOC), catalase (CAT), malondialdehyde (MDA), glutathione (GSH), oxidized glutathione disulfide (GSSG), and nitric oxide (NO).

### 2.7. Real-Time Quantitative PCR of Tight Junction Proteins

The relative mRNA expression of Occludin, Claudin-1 and ZO-1 in the lower part of the jejunum was detected using real-time quantitative PCR (RT-qPCR). The total RNA was extracted using a TRIzol kit obtained from Thermo Fisher (Shanghai, China), and the extracted RNA was reversely transcribed according to the instructions of the reverse transcription kit provided by Thermo Fisher (Shanghai, China). Then, the reverse transcriptional cDNA was diluted 10-fold for real-time quantitative PCR detection. The reaction system was 10 μL mixed with 0.3 μL forward primer, 0.3 μL reverse primer (Sangon Biotech, Shanghai, China), 0.4 μL cDNA, and 5 μL SYBR Green Supermix (Bio-rad, Shanghai, China), and the reaction conditions were as follows: 95 °C 5 min, 95 °C 10 s, 60 °C 10 s, 72 °C 30 s. The reaction was carried out in 40 cycles and a fluorescence signal was collected at the end of each cycle. Finally, the relative expression levels of each gene were calculated using the 2^−ΔΔCt^ method by using β-actin as the reference gene (Appendix A).

### 2.8. Short-Chain Fatty Acids (SCFAs) Quantification

The concentrations of SCFAs were determined by gas chromatography-mass spectrometry. A volume of 1 g stool mixed with 1 mL methanol was whirled for 30 s and centrifuged for 10 min (12,000× *g*, 4 °C). Afterward, 1 mL supernatant was added into 0.2 mL 25% metaphosphoric acid, left to stand at 4 °C for 30 min, and centrifuged for 10 min (12,000× *g*, 4 °C). Subsequently, 1 mL of the sample was automatically injected into the machine for measurement.

### 2.9. Gut Microbiota Profiling of 16S rDNA Sequencing

The total genomic DNA of fecal bacteria was extracted using the TGuide S96 Magnetic Soil/Stool DNA Kit obtained from Tiagen Biochemical Technology (Beijing, China). The extracted DNA was used for the PCR amplification of the V4 hypervariable regions of 16S rRNA genes, which were accomplished by primers 515F and 806R (515F, 5′-GTGYCAGCMGCCGCGGTAA-3′; 806R, 5′-GGACTACNVGGGTWTCTAAT-3′). Afterward, Pair-End was used to establish small fragment libraries for a sequencing procedure that was mainly processed on Illumina Novaseq 6000 platform (Illumina, San Diego, CA, USA) at Biomarker Technologies Co, Ltd. (Beijing, China). 

The Quantitative Insights segment of the Microbial Ecology software package (version 2021.6) was used for 16S raw data analysis [29]. The reads obtained after the sequencing result data were evaluated and optimized, with USEARCH being used to cluster reads with a similarity above 97.0% to generate operational taxonomic units (OTUs) that could be shown in a Venn diagram. To ensure the authenticity and accuracy of the sequencing results, KRONA was used to annotate the species and QIIME2 software (version 2021.11) (DADA2 denoise analysis) was used to evaluate the alpha diversity index of the samples, which mainly included Chao1, Ace, Shannon, Simpson, Coverage, and PD_whole_tree. The corresponding rarefaction curve was drawn using the R language package (version 4.3.1) and differential analysis was processed by one-way ANOVA, using Graphpad8.0.2 software. In addition, beta diversity analysis was performed by using QIIME software (version 2021.11) based on the Weighted Unifrac statistic algorithm to calculate the distance between the samples. The matrix distance was processed via analysis of similarities (ANOSIM) and principal coordinate analysis (PCoA) using the R language package (version 4.3.1). In addition, we used Greengene as the reference database and a Bayesian Classifier to execute taxonomic annotation on the feature sequence to obtain the species classification information corresponding to each feature. QIIME2 was then applied to obtain the abundance of each species in the samples; finally, a histogram of species distribution was generated using the R language package (version 4.3.1). Differential analysis among the groups of species abundance was processed by one-way ANOVA using Graphpad8.0.2 software.

### 2.10. Function Prediction and Correlation Analysis

PICRUSt2 was used to perform species annotation of feature sequences based on a reference phylogenetic tree. Potential functions and functional genes in the samples were predicted based on the Integrated Microbial Genomes (IMG) database, which further revealed the differences in functions between the samples or groups. STAMP software (version 2.0.0) was used for data analysis and the output of graphical results. In addition, Spearman rank correlation analysis was used to analyze the correlation between fecal score, villus height and crypt depth ratio, AOD, antioxidant index, and the mRNA expression of tight junction proteins and SCFAs with differential bacteria. The results were generated using Graphpad8.0.2.

### 2.11. Data Analysis

The differential analysis among the experimental data was processed by one-way ANOVA and Duncan’s multiple comparison using Graphpad8.0.2 software, and the data results were presented as mean ± standard error. Significance was expressed as * *p* < 0.05, ** *p* < 0.01 and *** *p* < 0.001.

## 3. Results

### 3.1. BE and FBE Improved Growth Performance in Weaned Piglets

After 2 weeks of BE and FBE feeding, the results showed that the ADFI (*p* < 0.05) and ADG (*p* < 0.01) at d 8–14 were improved. In addition, the fecal scores at d 1–7 (*p* < 0.05) and d 8–14 (*p* < 0.05) decreased. There were similar improvements in ADFI (*p* < 0.05) and ADG (*p* < 0.05) at d 8–14 in the FBE group, as well as a decrease in the fecal score at d 8–14 (*p* < 0.05) but not d 1–7 (*p* > 0.05). There was no significant difference in BW at d 1, BW at d 7, BW at d 14, and F: G (Table 2). 

### 3.2. BE and FBE Ameliorated Jejunal Tissue Morphology and Intestinal Epithelial Cells Apoptosis in Weaned Piglets

As shown in Figure 1A, compared with the Ctrl group, the BE and FBE groups improved epithelium damage of the lamina propria and villus integrity. Moreover, the BE and FBE groups improved the villus height and the ratio of villus height to crypt depth (*p* < 0.05) and decreased AOD (*p* < 0.05), but both the BE and FBE groups showed no significant difference in crypt depth compared to the Ctrl group (Figure 1B–E). 

### 3.3. BE and FBE Alleviated the Redox Imbalance and Barrier Dysfunction of Jejunum in Weaned Piglets

As shown in Figure 2, compared with the Ctrl group, the contents of T-AOC (*p* < 0.01), GSH (*p* < 0.05), and CAT (*p* < 0.01) increased, but the GSSG (*p* < 0.001) and MDA (*p* < 0.05) contents decreased in the BE group. Similarly, the contents of T-AOC (*p* < 0.05) and CAT (*p* < 0.05) increased, and the GSSG (*p* < 0.001) and MDA (*p* < 0.05) contents decreased in the FBE group. In addition, compared with the Ctrl group, the relative mRNA expression of the tight junction proteins Occludin, Claudin-1, and ZO-1 (*p* < 0.01) increased in the BE group, but the FBE group only increased the mRNA expression of Occuldin (*p* < 0.01) without significant changes to Claudin-1 and ZO-1 (*p* > 0.05) (Figure 2G–I).

### 3.4. BE and FBE Altered the Diversity of Gut Microbiota

The 16S rDNA sequencing results showed that the rarefaction curves (Figure 3A) gradually flattened with the increasing numbers of sequences, and the OTUs did not increase significantly, indicating that the sequencing data were adequate at presenting most species in the sample for analysis. The number of features obtained from the Ctrl, BE, and FBE groups was 529, 542, and 508, respectively; each group had its own unique number of features (Figure 3B). 

An intragroup analysis of beta diversity based on principal coordinates (PCoA) analysis showed that the intestinal microbial diversity changed in both the BE and FBE groups but not in the Ctrl group, and the differences were more pronounced in the BE group (Figure 3C). Compared with the Ctrl group, alpha diversity analysis (Figure 3D–J) showed that the Feature, ACE, Chao1, and PD_whole_tree indices were improved (*p* < 0.05) in the BE group, but there were no significant differences in the Simpson, Shannon, and Coverage indices. However, compared with the Ctrl and BE groups, there was no significant difference in the α diversity index in the FBE group, except for a decrease in the ACE index (*p* < 0.05) between the FBE and BE groups. 

### 3.5. BE and FBE Altered the Composition and Abundance of Gut Microbiota

As shown in the histogram of species distribution, *Firmicutes* and *Bacteroides* were the main microbial communities in the intestinal tract of piglets at the phylum level, accounting for over 80% (Figure 4A). Compared with the Ctrl group, dietary supplementation with BE increased the relative abundance of *Firmicutes* (*p* < 0.0001), but decreased the relative abundance of *Bacteroidetes* (*p* < 0.05). The relative abundance of *Firmicutes* (*p* < 0.01) also increased, but there was no significant effect on the relative abundance of *Bacteroides* in the FBE group (Figure 4B,C).

At the genus level (Figure 4D), for the bacteria that could be cultured, dietary supplementation with BE improved the relative abundance of *Lactobacillus* (*p* < 0.05) and *Phascolarctobacterium* (*p* < 0.01), but decreased the relative abundance of *Parabacteroides* (*p* < 0.01), *Succinivibrio* (*p* < 0.01), and *Butyricicoccus* (*p* < 0.01). Compared with the Ctrl group, the FBE group also exhibited a decreased relative abundance of *Parabacteroides* and *Succinivibrio* (*p* < 0.01), but no significant difference in the relative abundance of *Lactobacillus*, *Phascolarctobacterium*, and *Succinivibrio* (Figure 4E–I).

At the species level, we sorted the bacteria that had not been cultured into an uncultured group and placed the bacteria with a relative abundance of less than 0.1% into a group of others so that a histogram of the relative abundance of 14 species was obtained (Figure 4J). Compared with the Ctrl group, dietary supplementation with BE decreased the relative abundance of *Butyricicoccus_pullicaecorum* (*p* < 0.05) and *Ruminococcus_gnavus* (*p* < 0.01), similarly, it decreased the relative abundance of *Ruminococcus_gnavus* (*p* < 0.05) but increased the relative abundance of *Eubacterium_biforme* and *Prevotella_copri* (*p* < 0.05) in the FBE group (Figure 4K–N). 

### 3.6. BE and FBE Altered Microbial Function and the SCFAs Concentrations

To further explore the functional changes caused by the intestinal microbiota, we used the Kyoto Encyclopedia of Genes and Genomes (KEGG) analysis based on PICRUSt2 and STAMP software (version 2.0.0); a total of nine functions were found to have changed significantly. Compared with the Ctrl group, dietary supplementation with BE and FBE similarly upregulated the functions of carbohydrate metabolism, membrane transport, and cellular community of prokaryotes. Meanwhile, the functions of amino acid metabolism, biosynthesis of other secondary metabolites, metabolism of cofactors and vitamins, and energy metabolism were down-regulated in the BE group. The folding, sorting and degradation, metabolism of cofactors and vitamins, and energy metabolism were down-regulated in the FBE group. In addition, compared with the BE group, dietary supplementation with FBE upregulated the function of the immune system, and down-regulated the functions of infectious diseases, transcription, folding, sorting and degradation, translation, and immune diseases (Figure 5A–C). Among the altered intestinal microbial functions, carbohydrate metabolism is the main pathway of SCFA production. Compared with the Ctrl group, the dietary supplementation with BE increased the concentrations of propionate (*p* < 0.05) and butyrate (*p* < 0.01) in jejunal metabolites as well as the concentrations of propionate (*p* < 0.05) and butyrate (*p* < 0.05) in colonic feces (Figure 6A–F). 

### 3.7. Correlation Analysis between the Detected Indices and Differential Microbiota 

The Spearman’s correlation analysis showed that *Firmicutes* were positively correlated with villus height to crypt depth ratio (*p* < 0.05), T-AOC (*p* < 0.05), GSH (*p* < 0.05), CAT (*p* < 0.05), Occludin (*p* < 0.01), ZO-1 (*p* < 0.05), jejunal propionate (*p* < 0.05), jejunal butyrate (*p* < 0.05), and colonic propionate (*p* < 0.05), but were negatively correlated with AOD (*p* < 0.01). However, *Bacteroidetes* were negatively correlated with T-AOC (*p* < 0.05), GSSH (*p* < 0.05), Claudin-1 (*p* < 0.05), ZO-1 (*p* < 0.05), and jejunal acetate (*p* < 0.05). At the genus level, *Lactobacillus* positively correlated with T-AOC (*p* < 0.05), GSH (*p* < 0.05), CAT (*p* < 0.01), Occludin (*p* < 0.01), Claudin-1 (*p* < 0.05), ZO-1 (*p* < 0.01), jejunal acetate (*p* < 0.05), and colonic butyrate (*p* < 0.05). *Parabacteroides* were positively correlated with AOD (*p* < 0.01) and GSSG (*p* < 0.01), but negatively correlated with villus height to crypt depth ratio (*p* < 0.01), Occludin (*p* < 0.05), colonic acetate (*p* < 0.05), and propionate (*p* < 0.05). *Succinivibrio* were also positively correlated with AOD (*p* < 0.01) and GSSG (*p* < 0.01), but were negatively correlated with villus height to crypt depth ratio (*p* < 0.01), CAT (*p* < 0.05), Claudin-1 (*p* < 0.05), jejunal acetate (*p* < 0.001), jejunal propionate (*p* < 0.05), jejunal butyrate (*p* < 0.05), and colonic butyrate (*p* < 0.01). *Butyricicoccus* were negatively correlated with T-AOC (*p* < 0.05), ZO-1 (*p* < 0.01), and jejunal acetate (*p* < 0.05), whereas *Phascolarctobacterium* did not correlate with these indices. At the species level, *Ruminococcus_gnavus* was positively correlated with AOD (*p* < 0.001) and GSSG (*p* < 0.05), but negatively correlated with villus height to crypt depth ratio (*p* < 0.05), jejunal propionate (*p* < 0.05), and colonic propionate (*p* < 0.05). *Eubacterium_biforme* was positively correlated with GSSG (*p* < 0.05) and MDA (*p* < 0.05). However, *Prevotella_copri* did not show a correlation with these indices (Figure 7A). 

In addition, jejunal acetate was positively correlated with NO (*p* < 0.05), jejunal propionate was negatively correlated with AOD (*p* < 0.05), and jejunal butyrate was positively correlated with villus height to crypt depth ratio (*p* < 0.05) and CAT (*p* < 0.01) but negatively correlated with AOD (*p* < 0.05) and MDA (*p* < 0.05). Colonic acetate was positively correlated with GSH (*p* < 0.05) and colonic propionate was negatively correlated with AOD (*p* < 0.05). Meanwhile, jejunal acetate had a potentially positive correlation with villus height to crypt depth ratio and GSH, but it was potentially negatively correlated with AOD and GSSG. Jejunal propionate had a potentially positively correlation with villus height to crypt depth ratio, and jejunal butyrate was potentially positively correlated with T-AOC, Claudin-1, and ZO-1 but was potentially negatively correlated with GSSG (*p* < 0.1). Colonic propionate and butyrate had a potentially positive correlation with Occludin, and colonic butyrate was potentially negatively correlated with AOD, GSSG, and MDA (*p* < 0.1), although, there was no correlation between colonic butyrate and these indices (Figure 7B). 

## 4. Discussion

BBR improves the growth performance of weaned piglets and EA attenuates diarrhea in castor-oil-induced mice [30,31,32]. In this study, it was demonstrated that the combination of BBR and EA in the diet of piglets could improve growth performance by increasing ADG and ADFI and attenuating diarrhea in early-weaned piglets; a similar effect was achieved by the transplantation of piglet fecal suspension from the BE group. These results indicated that dietary supplementation with BE improved growth performance by acting on gut microbiota in weaned piglets.

Intestinal status is a key factor affecting the health of piglets. Oxidative stress induced by weaning can damage the intestinal structure and function [33], seriously affecting the health of piglets. BBR and EA have been proposed to ameliorate the structural intestinal damage caused by oxidative stress by improving intestinal tissue tightness and integrity [17,34]. Our results also indicated that BE in the diet of weaned piglets could improve jejunum oxidative stress by increasing T-AOC, GSH, and CAT content, and inversely decreasing GSSG and MDA content [35]. BE also increased the villus height to crypt depth ratio, increased the expressions of tight junction proteins between intestinal epithelial cells, and decreased the apoptosis of intestinal epithelial cells, thereby alleviating intestinal damage.

In recent years, an increasing number of studies have shown that BBR and EA exert important effects on the composition and function of gut microbiota [36,37]. In this study, we found that growth performance, intestinal morphology, intestinal epithelial cell apoptosis, and antioxidant capacities in the FBE group were similar to those in the BE group, suggesting that BE may affect the intestinal microflora in weaned piglets. To further explore the effects of BE on microbiota, 16S rDNA high-throughput sequencing was used. The results of the β diversity analysis showed that BE changed the composition of gut microbiota, and the FBE group was more inclined to the gut microbiota composition of the BE group compared with the Ctrl group. The results of the α diversity analysis showed there were significant differences in the Feature, ACE, and Chao1 indices between the Ctrl and BE groups, but the difference between the BE and FBE groups was not significant. Therefore, the results of the α and β diversity analyses indicated that the gut microbiota composition of the FBE group was more similar to the BE group compared with the Ctrl group. The similarity of the BE and FBE groups in regarding gut microbiota may lead to them exerting similar effects in terms of improving growth performance and intestinal damage. In addition, a further analysis of microbial abundance showed that piglets fed with BE had significantly increased *Firmicutes*, and an inversely decreased abundance of *Bacteroidetes* at the phylum level. The *Firmicutes* and *Bacteroidetes*, the main bacteria in the intestinal tract, play an important role in the intestinal stability and health of the host, and the ratio of *Firmicutes* to *Bacteroidetes* (F: B) is also an important biomarker of intestinal disorders [38,39]. A previous study showed that a decrease in F: B could improve obesity symptoms in mice and humans [40], thus, the increase in intestinal F: B in weaned piglets may have been related to the increase in ADG and ADFI in this experiment. At the genus level, piglets fed BE showed improved relative abundances of *Lactobacillus* and *Phascolarctobacteriu*, but decreased relative abundances of *Parabacteroides*, *Succinivibrio*, and *Butyricicoccus*. As one of the most dominant genera in the intestinal tract of piglets [41], *Lactobacillus* can resist the invasion of pathogens, strengthen intestinal barrier function, and improve immunity. Its metabolite, lactic acid, also plays an important role in the health of piglets [42]. Ikeyama et al. [43] showed that *Phascolarctobacteriu* can convert succinate to propionate, and the decrease in succinate could resist the colonization of the pathogenic bacteria *Clostridium difficile* in the intestinal tract, thus preventing Clostridium difficile infection [44]. Overall, most studies have revealed that *Lactobacillus* and *Phascolarctobacteriu* are beneficial to their hosts. Although BE increased the relative abundance of the SCFA-producing bacteria, *Lactobacillus* and *Phascolarctobacteriu* [45,46], it decreased the relative abundance of *Parapulobacters*, *Succinovibrio*, and *Butyricicoccus*, which also produce SCFAs. However, *Butyricicoccus* adversely affects the intestinal barrier by decreasing the expression of the tight junction Claudin-1 [47]. It was probable that BE could increase SCFAs by facilitating the colonization of the main dominant bacteria, *Lactobacillus* and *Phascolarctobacteriu*; thus, the relative decrease in the small proportion of *Parasulobacter*, *Succinovibrio*, and *Butyricicoccus* had no significant effect on the increase in SCFAs. At the species level, the piglets fed BE showed a significantly decreased relative abundance of the butyrate-producing bacterium *Butyricicoccus_pullicaecorum* [48], an anti-inflammatory bacterium that can treat inflammatory bowel disease [49]. In addition, BE decreased the relative abundance of the inflammatory bacterium *Ruminococcus_gnavus* [50,51]. However, the present study did not find that BE increased the abundance of relevant beneficial bacteria at the species level, perhaps because they were classified as uncultured or others (<0.1%). In addition, compared with the Ctrl group, the piglets fed FBE also showed an increased abundance of *Firmicutes*, and a decreased abundance of *Parapulobacters*, *Succinovibrio*, and *Ruminococcus_gnavus*, similar to the BE group. Respectively, FBE increased the abundance of *Eubacterium_biforme* and *Prevotella_copri*. *Eubacterium_biforme* is known as a SCFA-producer [52] and *Prevotella_copri* is considered an immune-relevant gut microbiota usually associated with a healthy lifestyle [53].

Functional analysis had shown that the dietary supplementation with BE significantly upregulated the metabolism of microbial carbohydrates, the main pathway for SCFA production [54]. SCFAs mainly include acetate, propionate, and butyrate, and our study found that BE increased the concentrations of propionate and butyrate in the jejunum and colon, which are produced mainly by *Firmicutes* and generally thought to benefit the health of the host [54,55]. SCFAs can provide energy for intestinal epithelial cells to facilitate their proliferation and differentiation, and the different concentrations of SCFAs have different effects on the epithelial cells of the intestinal villus and the crypt stem cells [56,57]. In combination with these studies, dietary supplementation with BE increased the villus height to crypt depth ratio and decreased the apoptosis of epithelial cells by increasing the concentrations of SCFAs in weaned piglets. In addition, SCFAs have a potential antioxidant capacity, and there are studies suggesting that sodium butyrate can improve the antioxidant capacity of pre-weaned calves [58] and increase the antioxidant capacity of carrageenan (CAR)-induced inflammatory mice by down-regulating the NF-κB pathway and up-regulating the activity of antioxidant enzymes [59]. Consistent with our results, the increase in intestinal butyrate concentration was positively correlated with antioxidant indices and negatively correlated with the content of the oxidative metabolite MDA. This indicated that dietary supplementation with BE improved the antioxidant capacity and ameliorated oxidative stress by increasing the butyrate concentration in weaned piglets. SCFAs also play an important role in protecting the integrity of the intestinal epithelial barrier, butyrate, in particular, has been shown to enhance the intestinal barrier by increasing the expression of Occludin, Claudin-1, and ZO-1 in human colonic Caco-2 and T84 cells and small intestine porcine IPEC-J2 cells [60,61,62]. This was consistent with the finding that the expression of tight junction proteins was potentially positively correlated with the butyrate concentration in this experiment. This indicated that dietary supplementation with BE probably enhanced intestinal epithelial barrier function by increasing the butyrate concentration in weaned piglets. Taken together, dietary supplementation with BE could attenuate intestinal oxidative stress and barrier dysfunction in weaned piglets by increasing SCFA concentrations.

## 5. Conclusions

Dietary supplementation with BBR and EA could improve growth performance and intestinal damage in weaned piglets by changing the composition of gut microbiota, such as increasing the ratio of *Firmicutes* and *Bacteroidetes* (F:B) as well as the abundance of *Lactobacillus* and *Phascolarctobacteriu*. Improvement in growth performance and intestinal damage were also associated with elevated levels of fecal SCFAs. Therefore, BE could be used as an effective additive in the diet of weaned piglets to improve their growth performance and intestinal damage. However, the underlying mechanism between BE and the differential microbiota warrants further investigation.

## Figures and Tables

**Figure 1 microorganisms-11-01254-f001:**
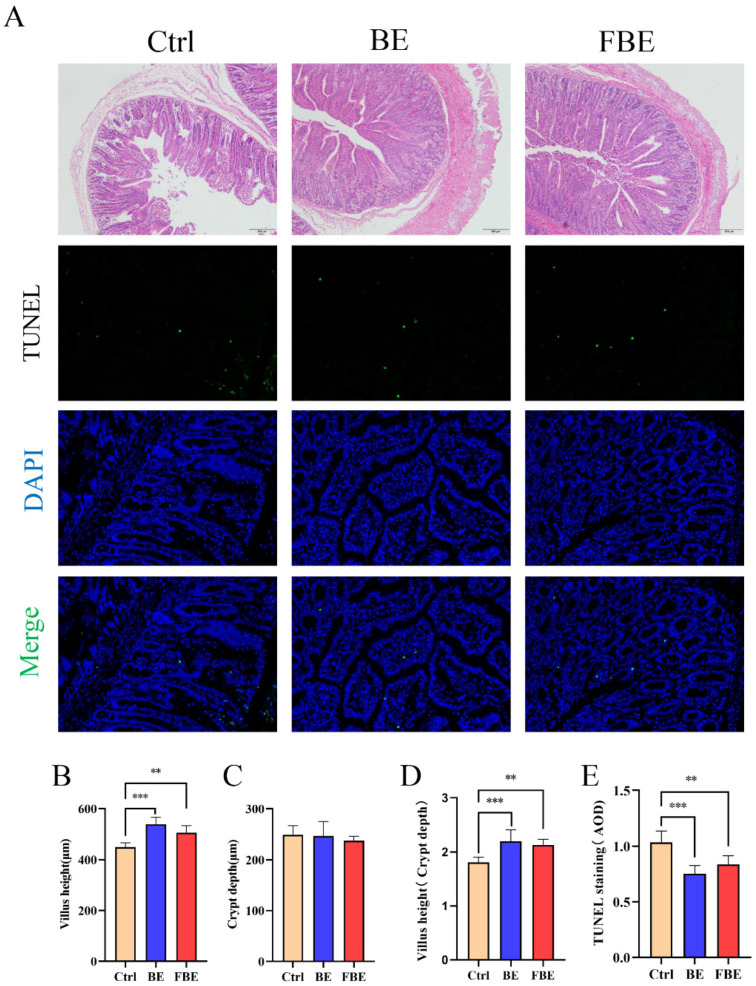
Jejunum tissue morphology and intestinal epithelial cell apoptosis of the Ctrl, BE, and FBE groups. Observations of HE (scale bar 200 μm) and TUNEL staining (scale bar 100 μm) (**A**). Villus height (**B**). Crypt depth (**C**). Villus height to crypt depth ratio (**D**). The AOD (green) of TUNEL staining (**E**). Ctrl group, piglets were fed basal diet; BE group, piglets were fed basal diet supplemented with Berberine and ellagic acid; FBE group, piglets were fed basal diet and received the fecal microbiota transplantation from piglets in BE group. HE, Hematoxylin-eosin; TUNEL, terminal deoxynucleotide transferase d-UTP terminal labeling; AOD, average optical density. Data were presented as Mean ± SEM and significance was indicated: ** *p* < 0.01, and *** *p* < 0.001.

**Figure 2 microorganisms-11-01254-f002:**
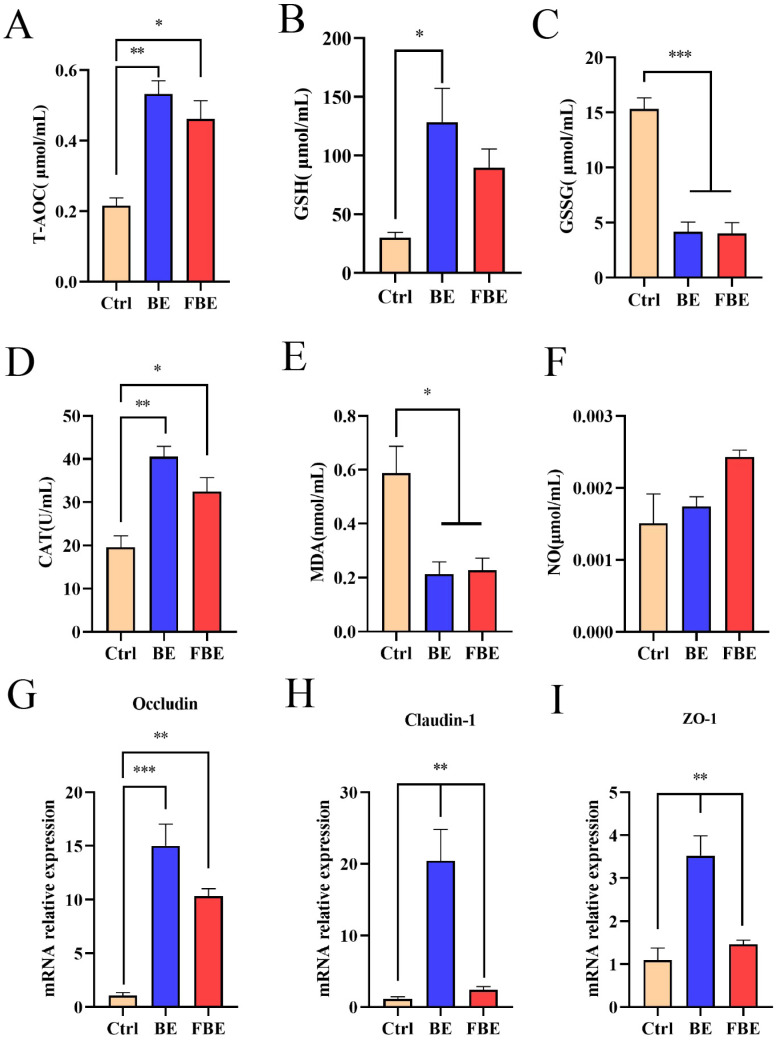
Antioxidant indices and mRNA expressions of tight junction proteins of the Ctrl, BE, and FBE groups. Anti-oxidative indices: T-AOC (**A**); GSH (**B**); GSSG (**C**); CAT (**D**); MDA (**E**); and NO (**F**). Tight junction proteins: Occludin (**G**); Claudin (**H**); and ZO-1 (**I**). Ctrl group, piglets were fed basal diet; BE group, piglets were fed basal diet supplemented with Berberine and ellagic acid; FBE group, piglets were fed basal diet and received the fecal microbiota transplantation from piglets in BE group. T-AOC, total antioxidant capacity; GSH, glutathione; GSSG, oxidized glutathione disulfide; CAT, catalase; MDA, malondialdehyde; NO, nitric oxide. Data were presented as Mean ± SEM and significance was indicated: * *p* < 0.05, ** *p* < 0.01, and *** *p* < 0.001.

**Figure 3 microorganisms-11-01254-f003:**
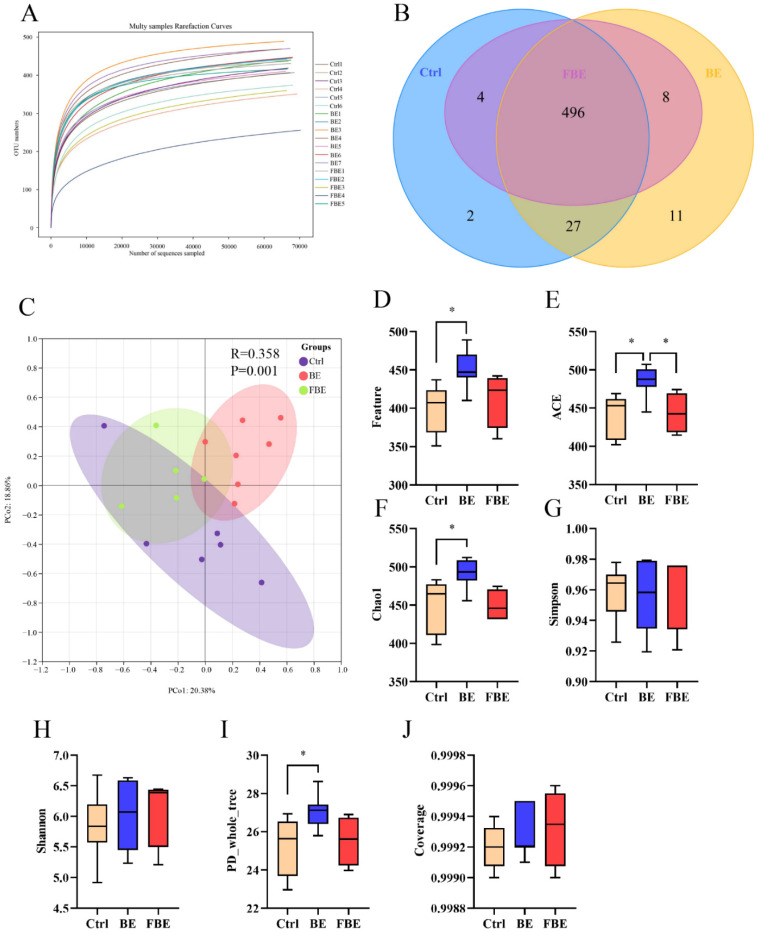
Diversity of gut microbiota of the Ctrl, BE, and FBE groups. Multi samples Rarefaction Curves (**A**) and Venn diagram (**B**) of numbers in each group. Beta diversity analysis based on principal coordinate analysis (PCoA) (**C**). Alpha diversity indices: Feature (**D**); ACE (**E**); Chao1 (**F**); Simpson (**G**); Shannon (**H**); PD_whole_tree (**I**); and Coverage (**J**). Ctrl group, piglets were fed basal diet; BE group, piglets were fed basal diet supplemented with Berberine and ellagic acid; FBE group, piglets were fed basal diet and received the fecal microbiota transplantation from piglets in BE group. Data were presented as Mean ± SEM and significance was indicated: * *p* < 0.05.

**Figure 4 microorganisms-11-01254-f004:**
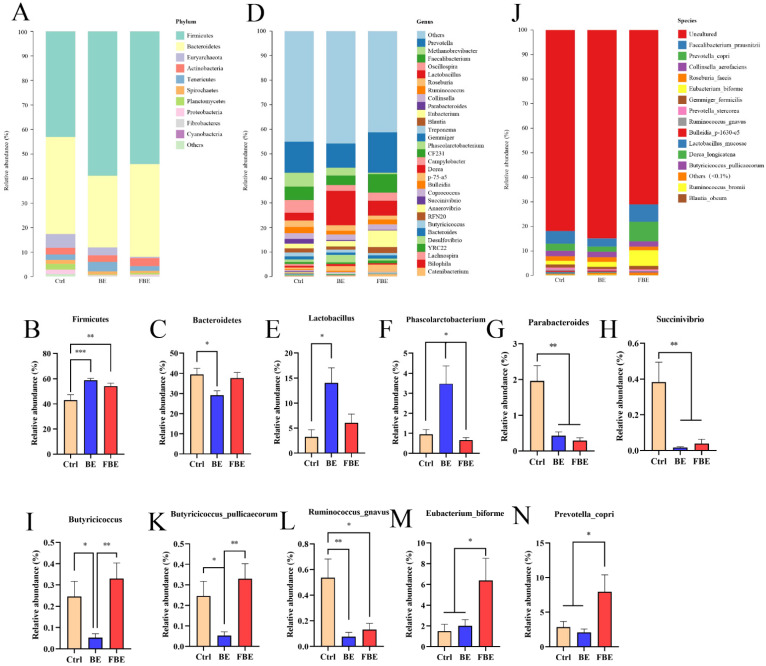
Composition of gut microbiota of the Ctrl, BE, and FBE groups. The microbial composition at phylum level (**A**). The relative abundance of differential bacteria at phylum level (**B**,**C**). The microbial composition at genus level (**D**). The relative abundance of differential bacteria at genus level (**E**–**I**). The microbial composition at species level (**J**). The relative abundance of differential bacteria at species level (**K**–**N**). Ctrl group, piglets were fed basal diet; BE group, piglets were fed basal diet supplemented with Berberine and ellagic acid; FBE group, piglets were fed basal diet and received the fecal microbiota transplantation from piglets in BE group. Data were presented as Mean ± SEM and significance was indicated: * *p* < 0.05, ** *p* < 0.01, and *** *p* < 0.001.

**Figure 5 microorganisms-11-01254-f005:**
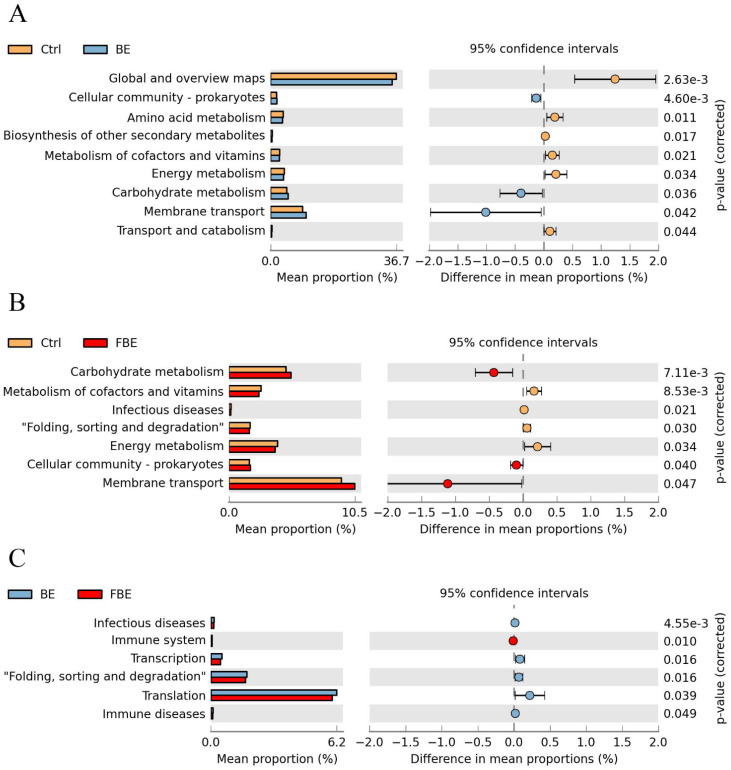
Microbial function of the Ctrl, BE, and FBE groups. The differential functional pathways between Ctrl and BE groups (**A**). The differential functional pathways between Ctrl and FBE groups (**B**). The differential functional pathways between BE and FBE groups (**C**). Ctrl group, piglets were fed basal diet; BE group, piglets were fed basal diet supplemented with Berberine and ellagic acid; FBE group, piglets were fed basal diet and received the fecal microbiota transplantation from piglets in BE group. The significance was presented as *p* < 0.05.

**Figure 6 microorganisms-11-01254-f006:**
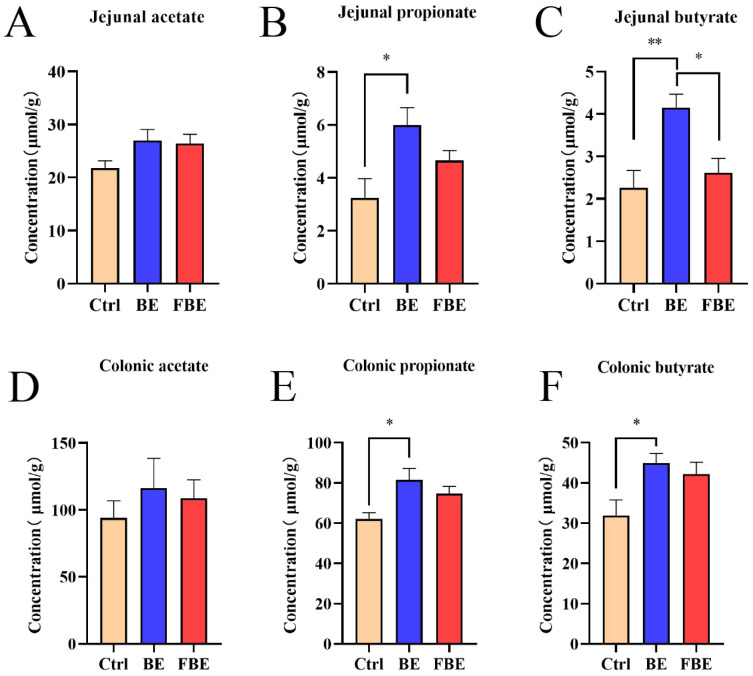
The concentrations of short-chain fatty acids (SCFAs) in jejunal and colonic content. The concentrations of jejunal SCFAs: acetate (**A**), propionate (**B**), and butyrate (**C**). The concentrations of colonic SCFAs: acetate (**D**), propionate (**E**), and butyrate (**F**). Ctrl group, piglets were fed basal diet; BE group, piglets were fed basal diet supplemented with Berberine and ellagic acid; FBE group, piglets were fed basal diet and received the fecal microbiota transplantation from piglets in BE group. Data were presented as Mean ± SEM and significance was indicated: * *p* < 0.05 and ** *p* < 0.01 represented potential significance.

**Figure 7 microorganisms-11-01254-f007:**
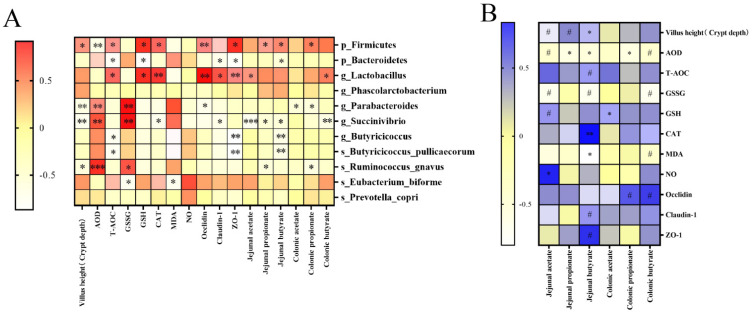
Correlation analysis between detected indices and differential microbiota. The correlation between detected indices and differential bacteria (**A**), the red and yellow colors represented positive and negative correlation, respectively. The correlation between short fatty acids (SCFAs) and detected indices (**B**), the blue and yellow colors represented positive and negative correlation, respectively. *P*, phylum; g, genus, s, species. The significance was indicated: * *p* < 0.05, ** *p* < 0.01, *** *p* < 0.001, and ^#^
*p* < 0.1 represented potential significance.

**Table 1 microorganisms-11-01254-t001:** Ingredients and nutrient levels of the basal diets (%).

Ingredients	Content	Nutrient Levels	Content
Corn	20.02	Dry matter	88.57
Soybean meal	8	Digestible energy (MJ/kg)	14.35
Extruded-soybean	10	Crude protein	19.01
Expended maize	35.00	Calcium	0.81
Fermented soybean meal	7.00	Total phosphorus	0.58
Soybean oil	1	Available phosphorus	0.42
Sucrose	3.00		
50% intestinal membrane protein powder	3.00		
Whey powder	6.00		
Fish meal	3.00		
Lysine	0.45		
Methionine	0.15		
Threonine	0.15		
Tryptophan	0.03		
Choline chloride	0.10		
Limestone	0.90		
Ca_2_PO_4_	0.90		
NaCl	0.30		
Vitamin–mineral premix ^1^	1		
Total	100		

Note: ^1^ The premix provides per kilogram of ration: Vitamin A: 15,500 IU; Vitamin D3: 3000 IU; Vitamin E: 40 mg; Vitamin K: 1 mg; Vitamin B1: 4.5 mg; Vitamin B2: 10.5 mg; Vitamin B6: 7 mg; Vitamin B12: 0.04 mg; Folic Acid: 2.0 mg; Niacinamide: 45 mg; D-Biotin: 0.3 mg; Iron, 100 mg; Copper, 6 mg; Manganese, 20 mg; Zinc, 90 mg; Iodine, 0.14 mg; Selenium, 0.30 mg.

**Table 2 microorganisms-11-01254-t002:** The growth performance of Ctrl, BE, and FBE groups.

Item	Ctrl Group	BE Group	FBE Group
Day 1 to 7			
BW at d 1, kg	6.08 ± 0.66	6.17 ± 0.41	6.31 ± 0.34
BW at d 7, kg	6.15 ± 0.66	6.25 ± 0.40	6.38 ± 0.34
ADG, g	66.69 ± 3.56	76.47 ± 0.95	72.40 ± 1.06
ADFI, g	145.02 ± 15.72	154.67 ± 5.22	147.79 ± 12.34
F: G	2.16 ± 0.15	2.01 ± 0.08	1.97 ± 0.15
Fecal score	3.00 ± 0.29 ^a^	1.88 ± 0.22 ^b^	2.38 ± 0.20 ^ab^
Day 8 to 14			
BW at d 14, kg	6.22 ± 0.66	6.34 ± 0.41	6.47 ± 0.35
ADG, g	70.27 ± 4.02 ^a^	93.34 ± 3.00 ^b^	83.56 ± 3.74 ^b^
ADFI, g	176.42 ± 7.79 ^a^	228.21 ± 13.96 ^b^	215.73 ± 16.34 ^b^
F: G	2.53 ± 0.13	2.50 ± 0.08	2.72 ± 0.15
Fecal score	2.5 ± 0.26 ^a^	1.38 ± 0.20 ^b^	1.5 ± 0.17 ^b^

Ctrl group, piglets were fed basal diet; BE group, piglets were fed basal diet supplemented with Berberine and ellagic acid; FBE group, piglets were fed basal diet and received the fecal microbiota transplantation from piglets in BE group; BW, body weight. ADG, average daily gain; ADFI, average daily feed intake; F: G, ratio of feed to gain; d 1, the day of weaning; d 7, day 7 post-weaning; d 14, day 14 post-weaning. Data presented as mean ± SEM. ^a,b^ Within a row, values with a different superscript were significantly different (*p* < 0.05).

## Data Availability

Not applicable.

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
