# Peer review of "Dietary Berberine and Ellagic Acid Supplementation Improve Growth Performance and Intestinal Damage by Regulating the Structural Function of Gut Microbiota and SCFAs in Weaned Piglets"

_microorganisms, 2023, doi:10.3390/microorganisms11051254_

Round 1

Reviewer 1 Report

This manuscript shows that administration of BE (berberine and ellagic acid) and an extract of BE-fed pig feces to weaning pigs improves food intake, growth and diarrhea. At the same time, stool analysis indicates that the intestinal flora is altered by BE. The present results may contribute the livestock industry. Refer comments.

Major comments

1, Abbreviations are used a lot and are difficult to read. The letters in many figures are too small. I would like you to think a little more about readers, such as creating a table of abbreviations, when making a manuscript.

2, In this study, intestinal tissue analysis was performed only in the jejunum. In connection with diarrhea and fecal bacteria, analysis of large intestinal tissue seems to be essential. Why is there no analysis result of the colonic tissue?

3, The important finding of this manuscript is that the pigs fed with the fecal extract of the BE-fed pigs showed beneficial effects similar to those of the BE-fed piglets. It is necessary to analyze how much viable bacteria are contained in this stool extract.

4, Berberine and ellagic acid are thought to be poorly absorbed in the small intestine, and most of them may be enter the large intestine and be excreted in the feces. Fecal extracts likely contain significant amounts of berberine and ellagic acid. Wouldn't it be necessary to have data on the contents of these compounds in the stool extract, and also the content data of the degradation products of these compounds by intestinal bacteria (low-molecular-weight polyphenols, etc.)?

5, This is related to comment 2, authors should discuss how bacteria in the large intestine (in the stool) affect various parameters of the jejunum.

6, There seems to be a regional or national bias in the papers cited.

Individual comments

1, L.19: BBR and EA are both administered 20 mg/kg BW? 

2, L82: Were authors using both male and female piglets in this study? If so, did authors try to ensure that there is no gender imbalance between groups? Are there any gender differences in observed parameters in weaning pigs?

3, L102, L120-146: What part of the jejunum is used in this study? Please specify the location. This reviewer thinks there are differences in histology and expression of various proteins between the upper and lower parts of the jejunum.

4, L.216, 225: The authors refer to "barrier dysfunction", but is there evidence of impaired intestinal barrier function in control piglets?

5, L.289: What exactly does the term “cellular community of prokaryotes” refer to?

6, Title of Figure 7 is wrong.

Reviewer 2 Report

The objectives of the research were to explore whether the combination of BBR and EA in diets could ameliorate diarrhea and intestinal damage by regulating gut microbiota in the early weaned piglets. It is of great help to the livestock industry under the background of antibiotic free. There are many sentences that are grammatically not correct and therefore it is difficult for a reader to understand what you are trying to say. I have mentioned some but not all in the specific comments and I would strongly advice to ask a native English speaker to correct your manuscript before submitting a revised version. Furthermore, in the introduction and materials sections a lot of information are missing, such as why is the combination of BBR and EA important? And the description of why use FMT and its possible mechanism are also needed. …… Because of these reasons I suggest major revisions for this manuscript.

Major concerns

Introduction

I think it is essential to mention why the combination of BBR and EA is important, because they are already known to play important roles on many aspects when it is used alone. And sometimes the side effect is caused occasionally when the combination, the author should clarify this.

It is abrupt that the use of FMT in the aim and the measurement of SCFAs. The descriptions of application of FMT should be mentioned ahead. Why is SCFAs? Also, what is the possible mechanism of gut microbiota on the treatment of diarrhea.

Materials and methods

21 piglets per group, 3 groups, and totally 63 piglets. Is the biological duplication unnecessary? What is the rearing condition for these piglets? 21 piglets for a pigsty? How big is the pigsty?

Important information for FMT is missing, and the details of proposal of FMT should be provided. Does it orally treat? How many days for the FMT treatment? Are donor piglets screened to be healthy before the FMT? Is the fecal sediment from one piglet or the mixture of 21 piglets of the BE group?

Fecal score is point variable, it is not correct to be calculated by ANOVA. The chi square may be appropriate. Thus, the description of percentage for each score instead of average is better.

Results

It is rare to analyze the species microbes because the tech of 16srDNA is not precise the determine the specie level. Metagenomics is needed to apply for the species microbes.

The legend of Figures is not accurate. It should be objective and readers could not the difference from the picture. Such as, Figure 1 should be “The growth performance of Cont, BE, and FBE groups”. Figure 2 should be “jejunum tissue morphology and intestinal epithelial cell apoptosis of Cont, BE, and FBE groups”. And the same as other figures.

How is the KEGG pathways between the comparison of BE and FBE?

The alpha diversity between the control and FBE group is not different, and similar result from the result of beta diversity is obtained that the microbiota of BE is much different from the control and BE group. Also, there is no difference of phylum and genus microbes as well as SCFAs between the two group.

Discussion

I don’t follow what was the evidence that the concentrations of propionate and butyrate in the jejunum and colon were produced mainly by Firmicutes? Is it from the correlation?

There is lack of the discussion for alpha and beta diversity of the gut microbiota. Besides, the effect of FMT should be also discussed in more details.

Specific concerns

Line 17, a total of.

Line 22, and reduced fecal scores

Line 23, please specify “it”

Line 25, while attenuate

Line 27-28, Interestingly, oral-administration of fecal microbiota suspension from piglets fed with BE had similar effects to BE group. From the results of alpha and beta diversities, and the abundance of phylum_ and genus microbes, I am not sure this conclusion is reasonable.

Line 22, 26, ADG, ADFI, T-AOC, GSH and CAT, please use the full name

Line 47, much attentions

Line 51, A study

Line 51-54, it is hard to learn this sentence “Study has shown that BBR prolongs the time to diarrhea in a dose-de-pendent manner and improve diarrhea-predominant irritable bowel syndrome (IBS-D) symptoms in mice and pre-treating intestinal epithelial cells with BBR enhances barrier function in pigs.”.

Line 59, A previous study

Line 147 SCFAs should be written as a full name when in the title.

Line 193-195 “These results indicated that supplementation of BE and FBE in the diet could improve the growth performance in weaned piglets” should be written in discussion section.

Line 202, there is lack of description for Figure 2A.

Line 206-209, 223-225, 246-247, 298-299 should be written in discussion.

Line 294-296, delete this sentence “Also, the concentrations of SCFAs in BE group and FBE group were increased with the up-regulation of carbohydrate metabolism.”.

Line 305, please specify “Figure 7. Correlation analysis between detected indexes and differential microbiota”.

Line 356, it was demonstrated.

Line 365, improved intestinal tissue tightness.

Line 357-368, by increasing T-AOC, GSH and CAT content, and inversely decreasing GSSG and MDA content.

Line 368, please classify “it”.

Line 372-375, Please re-organize the sentence, “In this study, through preliminarily observation, the results of growth performance and intestinal damage in FBE group, we found it had similar beneffcial effects to BE group, suggesting that BE may affect the intestinal microffora in weaned piglets.”

Line 375-377, To further explore the effects of BE on microbiota, the use of 16S rDNA high-throughput sequencing was used, and the results of α and β diversity analysis showed that BE changed the composition of gut microbiota.

Line 377-382, In addition, Further analysis of microbial abundance showed that piglets fed with BE signiffcantly increased Firmicutes and inversely decreased Bacteroidetes abundance at phylum level. The Firmicutes and Bacteroidetes, as the main bacteria in the intestinal tract, play an important role in the intestinal stability and health of the host, and the ratio of the Firmicutes and Bacteroidetes (F: B) is also an important biomarker of intestinal disorder.

Line 382, A previous studies.

Lline 388-389, Lactobacillus can resist the invasion of pathogens, strengthen the intestinal barrier function and improve immunity,

Line 392, thus prevent.

Line 419-428, 428-435, please re-organize the following sentences by divided into several sentences. “In addition, SCFAs have potential antioxidant capacity, there are researches suggesting that adding sodium butyrate to the diet of pre-weaned calves can improve the antioxidant capacity of calves[52] and sodium butyrate increases the an-tioxidant capacity of carrageenan (CAR) -induced infammatory mice by down-regulating the NF-κB pathway and up-regulating the activity of antioxidant enzymes[53], consistent with our experiment, the increase of intestinal butyrate concentration was positively cor-related with antioxidant indexes and negatively correlated with oxidative metabolite MDA, it indicated that dietary supplementation of BE presumably improving the antiox-idant capacity and ameliorating oxidative stress through increasing butyrate concentra-tion in weaned piglets.:, and “SCFAs also play an important role in protecting the integrity of the intestinal epithelial barrier, particularly butyrate has been showed to enhance the intesti-nal barrier through increasing the expressions of Occludin, Claudin-1 and ZO-1 in human colonic Caco-2 and T84 cells and small intestine porcine IPEC-J2 cells[54-56], which was consistent with the result that the expressions of tight junction proteins and butyrate con-centration had a potentially positive correlation in this experiment, it indicated that the supplementation of BE in the diet was probably enhancing intestinal epithelial barrier function by increasing butyrate concentration in weaned piglets.”.

Line 442, please classify “it”.

Line 466, 474, 480, 484, 503, 506, 527, 529, 534, 539, 541, 545, 552, 565, 572, 575, 577, 582, 594, Please use the abbreviation of journal name.

Reviewer 3 Report

Author study Dietary berberine and ellagic acid supplementation improve growth performance and intestinal damage by regulating the structural function of gut microbiota and SCFAs in weaned piglets. The text is required to be reviewed by a professional editor. Information regarding to statistical analysis is insufficient to repeat the procedure, especially for relative abundant and alpha diversity analysis. A comprehensive discussion on the results of FBE is missing.  Experimental design, material preparations, and animal husbandry are insufficient.

Line 17: “and the mechanism is limited.”

This sentence is not clear.

Line 18: The treatment description is not clear.

Line 38: Please define the early weaning.

Line 82: Please provide housing conditions. How many pigs were housed in each pen?

What is the gender of the pigs used in this study? Was the average BW of pigs comparable between treatments at weaning?

Line 84: Please detail which pigs or pen feces were collected FMT. Were feces collected each day from BE group being used for FMT? Moreover, please specify how the fecal suspension was prepared.

Line 86: When were pigs killed to tissue sampling? What was the body weight ranking of pigs from the pen that was selected for slaughter?

Table 1 Please check the diet table.

Line 99 What is presyringe? Is this manufacturer’s product trading name?

If it is please provide the company information.  

Line 108 How many pigs of stool were collected to prepare fecal suspension?

Was fecal suspension prepared fresh daily? Were pigs received FMT on the first day?

Line 111: Please specify the housing system, type of feeders, number of pigs per pen, and size of the pen. Also, was the person who score diarrhea blinded for treatments? What is the basic unit of the diarrhea score? Each pig or each pen.

Line 152 please revise mL to ml for the whole text.

Line 158 What is 4v_b?

Line 162: please specify whether the deblur or DADA2 was used in QIIme2.

Line 183: Please specify fixed factors, random factors, and an experimental unit that is used for data analysis. The 16S data analysis statement is missing.

Line 189: was ADG at d 14 represent the ADG on d 14 or during d 0-14?

Does feed efficiency data available?

Line 222: Note that Caludin-1 and ZO-1 were not greater in FBE.

Line 240: Figure 1C Please provide the R and P value for beta diversity.

How were the significant level of alpha diversity indexes between treatments determined?

Figures 5 B to N please specify how the statistical analysis was performed.

Line 374 Results indicate that some responses from traits measured in fact are the same, which should be discussed.

Line 407: How is FBE different from Control and BE?

Round 2

Reviewer 1 Report

The manuscript has been revised correctly.

Reviewer 2 Report

The authors have revised most of the issues. However, the statistical analyse for the fecel score still is needed to be revised. Chi square is appropriate rather than one-way ANOVA for point variable.

Author Response

 The manuscript has received the extensive editing of English language and style

, please see the attachment.

Reviewer 3 Report

Line 18: The treatment description is not clear.

Response: Thanks for your suggestion. the treatment description has been changed to “Piglets in the Ctrl group were treated with basal diet and 2 ml saline orally, while those in the BE group were treated with basal diet supplemented with 10 mg/kg (BW) BBR, 10 mg/kg (BW) EA, and 2 ml saline orally. Piglets in the FBE group were treated with basal diet and 2 ml fecal microbiota suspension from the BE group orally, respectively, for 14 days.” (L18-22)

Q: was BBR and EA premixed with saline into a 2 ml mixture?

 What is the gender of the pigs used in this study? Was the average BW of pigs comparable between treatments at weaning?

Response: Thanks for your question. There were 4 male piglets and 3 female piglets each pen.

I have added the table of average BW into article. I added the average BW of piglets into table 2 (L217).

Q: It appears that the beginning BW of BE and FBE groups were heavier than the control. Note that weaning BW greatly impacts the outcome of growth performance.

Please check with Journal’s standard for the table. Ideally, a table or figure should be able to stand alone which means any abbreviation should be defined.

Please check the superscript on the fecal score during d 8-14.

Line 86: When were pigs killed to tissue sampling? What was the body weight ranking of pigs from the pen that was selected for slaughter?

Response: Thanks for your question. the piglets were killed at 15 days after weaning. Piglets weighed 6±1 kg were selected form the pen for slaughter.

Q: In this case, the pigs were selected specific for sampling instead of randomly as indicates in line 115.

Line 222: Note that Caludin-1 and ZO-1 were not greater in FBE.

Response: Thanks for your notice. Compared with the Ctrl group, FBE improved the mRNA expression of Occludin, but there was no significant difference in Caludin-1 and ZO-1. It would indicate that the FBE treatment could not achieve the exact same effects of expression on Claudin-1 and ZO-1 as BE treatment.

Q: again the Claudin-1 and ZO-1 from FBE were not greater than the control group. Thus, the statement in lines 288-290 is incorrect.

Line 240: Figure 1C Please provide the R and P value for beta diversity.

Response: Thanks for your question. the R value was 0.385 and P value was 0.001.

How were the significant level of alpha diversity indexes between treatments determined?

Response: Thanks for your question. differential analysis among groups on alpha diversity metrics was processed by one-way ANOVA (L188).

Q: which procedure of one-way ANOVA was used to analyze the microbiome results?
